# Determinants of enrollment in community based health insurance among Households in Tach-Armachiho Woreda, North Gondar, Ethiopia, 2019

**Muluken Genetu Chanie**[1]*, **Gojjam Eshetie Ewunetie**[2]

**1** School of Public Health, College of Medicine and Health Sciences, Wollo University, Addis Ababa, Ethiopia, **2** Department of Medical Laboratory Sciences, Denbya Primary Hospital, North Gondar, Amhara Regional State, Ethiopia

* mlkngnt@gmail.com

**Data Availability Statement:** All the data supporting the findings are within the paper. Additional detailed information and raw data are available from the corresponding author or the

## Abstract

### Background

Recently in Ethiopia, there is an increasing movement to implement community based health insurance scheme as integral part of health care financing and remarkable movements has resulted in the spread of the scheme in different parts of the country. Despite such increasing effort, recent empirical evidence shows enrolment has remained low. To identify determinants of enrollment in community based health insurance among households in Tach-Armachiho Woreda, North Gondar, Ethiopia, 2019.

### Methods

A community based unmatched case control study was conducted Tach-Armachiho Woreda from March to May 2019 among 262 participants (88 cases and 174 controls with case control ratio of 1:2). Study subjects were selected using multi-stage sampling technique. Data were collected using a pretested, structured interviewer administered questioner. Data were entered to Epi-info 7 and exported to SPSS version 20 for analysis. Bivariable and multivariable logistic regression model were used to see the determinants of enrollment in community based health insurance. Adjusted odds ratio with 95% CI at p-value <0.05 in multivariable logistics regression analysis factors were identified as statistically significantly associated.

### Result

Female headed households (AOR = 2.79, 95% CI = 1.16, 6.69), Increase in Age (AOR = 1.09, 95% CI = 1.05, 1.13) and negative perception towards community based health insurance (AOR = 0.062, 95% CI = .030, .128) were found to be significant predictors.

### Conclusion

This study provides evidence that the decision to enroll in the scheme is shaped by age and a combination of household head sex and perception towards community based health

Institutional Review Board of University of Gondar (nigussuworku29@gmail.com) on reasonable request.

**Funding:** The finance of the research was funded by University of Gondar.

**Competing interests:** No authors have competing interests.

insurance. Implementers aimed at enhancing enrolment ought to act on the bases of this findings.

## 1. Background

Over the past two decades, many low-income and middle-income countries (LMICs) have found it progressively more difficult to maintain sufficient financing for healthcare services in an equitable way [1, 2]. This distressing scenario triggered WHO and other international body to propose an alternative approach in the late 1990s; thereby, various forms of community based health care financing have been emerged [3, 4]. Community based health insurance (CBHI) schemes are classically risk-pooling approach that tries to spread health costs across households with different health profiles to ensure better access and enables cross subsidies from rich to poor populations [5].

In June 2011, the Government of Ethiopia rolled out a pilot basis CBHI scheme in 13 woredas of Amhara, Oromia, Southern Nations Nationalities and Peoples (SNNP), and Tigray regional states [6, 7]. Since its establishment, there has been increasing movement to support and spread pilot schemes in different parts of the country. For instance, by the end of June 2017, the number of scheme has grown in to 377 pilot woredas [8]. Despite increasing support and spread of CBHI as noted before, recent empirical evidence shows enrolment has remained low across the scheme in being implemented areas [7, 9] indicating that CBHI has continued to fail to reach satisfactory levels of participation amongst targeted population. As of June 2017 the federal ministry of health (FMOH) annual report shows, the overall enrollment rate in the pilot scheme was 36% [9].

One possible explanation for low scheme uptake is that, household or individual level characteristics combined with how one perceives CBHI scheme in the dimension of its effectiveness in meeting health care needs; quality of care; trust and provider competency. As it was reported from the endowment effect and status quo bias complicated for individuals the decision to insure particularly in areas where insurance is a new concept and illiteracy rates are high, availability of health care facilities; episode of chronic illness in the household and an understanding of the product [10, 11].

Furthermore, scheme related factors such as affordability; benefit packages and payment mechanisms also affect scheme uptake [3, 5, 12]. However uptake of the Ethiopian CBHI program reveals the opposite, with the poorest quintile providing the largest share of CBHI beneficiaries and there is general agreement that the defined benefit package is adequate [7, 13].

While there are concerns about scheme uptake and suggested factors to the problem, scientific evaluation of the factors affecting the decision to enroll in CBHI the scheme in being implemented areas are still very scarce [7].

As to the researcher knowledge, in Ethiopia a few studies were conducted on subjects related to CBHI which mainly focused on factors affecting willingness to join (WTJ) [6, 14, 15] and impact of CBHI and scheme performance [7, 13]. Currently, Ethiopia has begun establishing a comprehensive and sustainable risk protection system with health care financing mechanisms adapted to our country's needs so as to improving financial access to health care services; improving quality of health care service and increase resource mobilization in the health sector through CBHI. However, the objective is, there is little attention has been paid to understand factors affecting uptake of CBHI, this can partly be attributed low enrollment in CBHI. Therefore, the subject should be studied and it provides information on factors

affecting uptake of CBHI so as to design interventions to increase scheme uptake. There was no study that documented on determinants of enrollment in CBHI in Ethiopia in general and in Amhara Regional State in particular.

## 2. Methods

### 2.1. Study design and setting

A community based unmatched case control study was conducted in Tach-Armachiho district from March 28th to May 15th, 2019. Tach-Armachiho is found in North Gondar 68km from Gondar city, Amara Region, northwest Ethiopia located 793 km from Addis Ababa on the main road to Humera. The District has an estimated 224,842 inhabitants with in thirty-seven kebeles. The district has seven health centers, forty-three health posts. The district has been seen implementing CBHI program since 2016. As of March 2018 district health office report, from the total of 45,237 eligible households, 11761 households were enrolled in the scheme among these 870 households were indigents [9].

### 2.2. Population

The source population of the study was all population of Tach-Armachiho District. The study populations were all population in selected kebeles of the district, who is officially registered for the year of 2018 and non-enrolled households who lived for more than six months in the study kebeles.

### 2.3. Sample size determination and sampling technique

The sample size was determined using Epi-info version 7.0.8.3 and based on the following assumptions: Precision 5% at 95% confidence level, power of 80%. The ratio of controls to cases (r) = 2, OR = 3.47 and $P_2$ = 24.4% [12]. Considering the design effect of 2, and possible non-response rate 15%, a sample size of 262 (88- cases and 174- controls) households were included in the study. The study participants were drawn using a multi-stage sampling technique. In the first stage, 11 Kebeles were selected randomly (lottery method) out of thirty-seven kebeles considering the capacity of conducting the research. The second stage involved the selection of households from eleven kebeles. List of households for cases and controls obtained from each kebele's administration household record list which was used as a sampling frame. The sample size was proportionally allocated for selected kebeles based on each kebele's number of households. Then simple random sampling (lottery method) was employed to select the cases and controls by taking their list as a frame and labeling continuous numbering.

### 2.4. Operational definitions

Availability of health facility: in this study implies that existence of governmental health facility within a 30 minute walking distance.

Chronic illness experience: it is households or memeber of households who get an illness lasting more than six months preceding the data collection period.

Perception: A five-point Likert Scale ranging from '1 strongly disagree' to '5 strongly agree' was used for respondents to express their opinions on four perception tools. In order to rank them according to their relative importance, first the minimum attainable score was determined. Then the minimum attainable score (4) multiplied by the corresponding grades in the scales (1, 2, 3, 4 and 5) to get the maximum attainable scores for each grade in the scale was

determined. Then median score was calculated and graded as positive perception (for values above the median score) and negative perception (for values less than median score).

## 2.5. Data collection tools and procedures

Data collection tools were prepared from reviewing literatures and manuals from different sources and adopted for this study accordingly with discussion and consultation of experts [11, 16, 17]. The survey tools were adopted from reviewing a couple of previous literatures. They were written in English language. They were paraphrased accordingly and are referenced above. No new tools were developed by the authors. But it was translated to local language (Amharic) by language experts for ease of communication and understanding. Before beginning the data collection procedure, the study subjects were identified from the record of the kebele administration household head list. The data was collected by trained data collectors using a pre-tested, structured and interviewer administered questionnaires. The questionnaire was developed in English and then translated into Amharic. To check for its consistency, the questionnaires were translated back to English by English language experts. The principal investigator was supervised the data collection process by checking completeness of the required type of data and correcting for errors at field.

## 2.6. Data processing and analysis

Data gathered through structured questioners were entered into Epi-info version 7.0.8.3 and exported to SPSS version 20 statistical software for analysis. The data were cleaned for inconsistencies and missing values. Bivariable and multivariable logistic regression model were used to see the determinants of enrollment in CBHI. To determine the effect of each independent variable on enrollment bivariable analysis was performed. Then all independent variables with p-value < 0.25 in the bivariable regression were fitted to multivariable logistic regression model. AOR with 95% CI at p-value <0.05 were used to declare significant association of determinants on enrolment of CBHIC among households of Tach-Armachiho district.

## 2.7. Data quality management

Prior to the actual data collection, pretest was carried on 5% (5 cases & 10 controlls) of the sample on similar population that are not part of the actual sample to make further adjustment (Households of those kebeles which were not selected for the actual study). In addition, training was given for 6 data collectors (BSc nurses) and two supervisors (health officers) to familiarize them with the questioner. Close supervision during data collection had been carried out by the principal investigator and supervisors and data was checked for completeness and consistency on spot as well.

## 2.8. Ethical consideration

Ethical clearance was taken from ethical review board of University of Gondar, College of Medicine and Health Sciences. Letter of permission to conduct the study was obtained from administrative office of Tach-Armachiho district. Written informed consent was obtained from participants before data collection. They were informed that participating in the study was voluntarily. The right to withdraw from the study at any moment during the interview was assured. No personal identifiers were used on data collection format. The recorded data were not accessed by a third person except the principal investigator, and was kept confidentially and anonymously.

## 3. Result

Two hundred sixty two households were interviewed resulting in an overall response rate of 100%. Among those interviewed 88 were those who enrolled for CBHI (cases) and 174 were those who do not enrolled (controls) for CBHI.

### 3.1. Socio-demographic characteristics of households

The ages of respondents were found to range from 27 to 75 years, with a mean age of 50.5 years (SD = ±10.26) for cases and it was ranged from 22 to 75 years, with a mean age of 40.8 (SD = ± 10.58) for controls. From the total respondents, 62 (70.5%) of cases and 144 (82.8%) of the controls were males. Majority of the study participants 61 (69.3%) of cases and 143 (82.2%) of controls were Muslims and all respondents belongs to Amara ethnicity. For cases, the mean family size was 4.49 (SD = ± 1.58) with a range of 1 to 11 family numbers and for controls, the mean family size was 4.53 (SD = ± 1.96) with a range of 1 to 10 family numbers.

Only 20 (11.5%) of the respondents in the control group had secondary/tertiary education, while 43 (48.9%) of the cases and 94 (54%) of controls never had formal education. More than 67 (78%) of cases and 136 (78%) of controls were married and a larger percentage of (63.4%) households spent less than 30 minutes to reach the nearby health institution. The mean time taken to reach to the nearby health institution on foot was estimated to be 53 minutes (SD = ± 49) for controls and 32 minutes (SD = ± 19) for cases. With regarding chronic illness, 92.0% of cases and 96.6% of controls had no chronic illness among their family members.

### 3.2. Perception of households towards CBHI scheme

Almost comparable numbers of cases (54.5%) and controls (63.2%) were agreed with the statement that CBHI management is trust worthy (Table 1).

As shown in Fig 1, the result indicates that data on perception scale scores were ranked with households having positive and negative perception towards CBHI scheme. Among participants who had good perception towards CBHI 76.14% were enrolled in the CBHI scheme (Fig 1).

### 3.3. Factors associated with uptake of CBHI among households

In the Bivariable analysis 10 variables were found significant at p-value <0.25 with 95% CI and they were fitted for further analysis in multivariable logistics regression model to control confounders and to ascertain the effect of each independent variables on the likelihood of CBHI enrolment. A final multivariable logistic regression analysis was performed to assess the impact of a number of factors on the likelihood that household heads to enroll in CBHI. The model contained four independent variables and as a whole explained between 38% (Cox & Snell R Square) and 52.8% (Nagelkerke R Square) of the variance in enrollment status and correctly classified 83.6% of cases.

The multivariable logistic regression result showed that female headed household were 2.79 at 95% CI: (1.16, 6.69), (p = 0.022) times more likely to enroll in CBHI than male headed households. With regard to the age of the household head, for an additional one year in age, the odds of enrolling is increased by a factor of 1.09 at 95% CI: (1.05, 1.13) (p = 0.0001) times. Concerning perception of households towards CBHI scheme it was found that, household heads who had negative perception towards CBHI scheme (CBHI not beneficial, not trusted, poor quality of care and poor staff performance) were 0.062 (95% CI: .030, .128)) (p = 0.000) times less likely to enroll than those household heads who don't think as such (Table 2).

**Table 1. Perception factors and Likert scale score of respondents for inrolment of household to CBHI in Tach-Armachiho district, North Gondar, Ethiopia, 2019.**

| Variables | Likert scale | Uptake of CBHI | |
|---|---|---|---|
| | | Yes n (%) | No n (%) |
| CBHI benefit packages are adequate enough to meet health care needs of your household. | SD | 10 (11.4%) | 67 (38.5%) |
| | D | 13 (14.8%) | 78 (44.8%) |
| | A | 37 (42%) | 24 (13.8%) |
| | SA | 28 (31.8%) | 5 (2.9%) |
| CBHI management is trust worthy. | SD | 5 (5.7%) | 42 (24.1%) |
| | D | 5 (5.7%) | 17 (9.8%) |
| | A | 48 (54.5%) | 110 (63.2%) |
| | SA | 30 (34.1%) | 5 (2.9%) |
| The quality of health care services is good (waiting time, availability of drugs, diagnostics) | SD | 10 (11.4%) | 101 (58%) |
| | D | 21 (23.9%) | 51 (29.3%) |
| | A | 38 (43.2%) | 17 (9.8%) |
| | SA | 19 (21.6%) | 5 (2.9%) |
| The provider makes a good diagnosis | SD | 13 (14.8%) | 97 (55.7%) |
| | D | 13 (14.8%) | 41 (23.6%) |
| | A | 37 (42%) | 30 (17.2%) |
| | SA | 25 (28.4%) | 6 (3.4%) |

SA: Strongly Agree; A: Agree; D: Disagree; SD: Strongly Disagree.

## 4. Discussion

The objective of this study was to identify determinants of enrollment in CBHI among households in Tach-Armachiho district. An analysis of data collected through structured questionnaire revealed that, there was no significant association between marital-status, family size, educational status, distance, religion, and chronic illness in line with uptake of CBHI.

One of the variables found to be significantly related to uptake of the scheme was sex of the household head. This study revealed that female-headed households were more likely to enroll as compared to male-headed households. This finding was consistent with studies done in Pilot Schemes across the country that revealed, female-headed households were found to be more likely to enroll as compared to male-headed households [7]. Similarly, studies conducted in Ghana, Mali and Senegal has shown that female-headed households were more likely to enroll in CBHI schemes [10, 18].

The fact that female-headed households were found to be more likely to enroll as compared to male-headed households might be different risk perceptions regarding disease as males were more risk takers than females.

On the contrary, studies conducted in Fogera Woreda, North West Ethiopia, Burkina Faso and Nigeria found that male-headed households were found to be more likely to enroll as

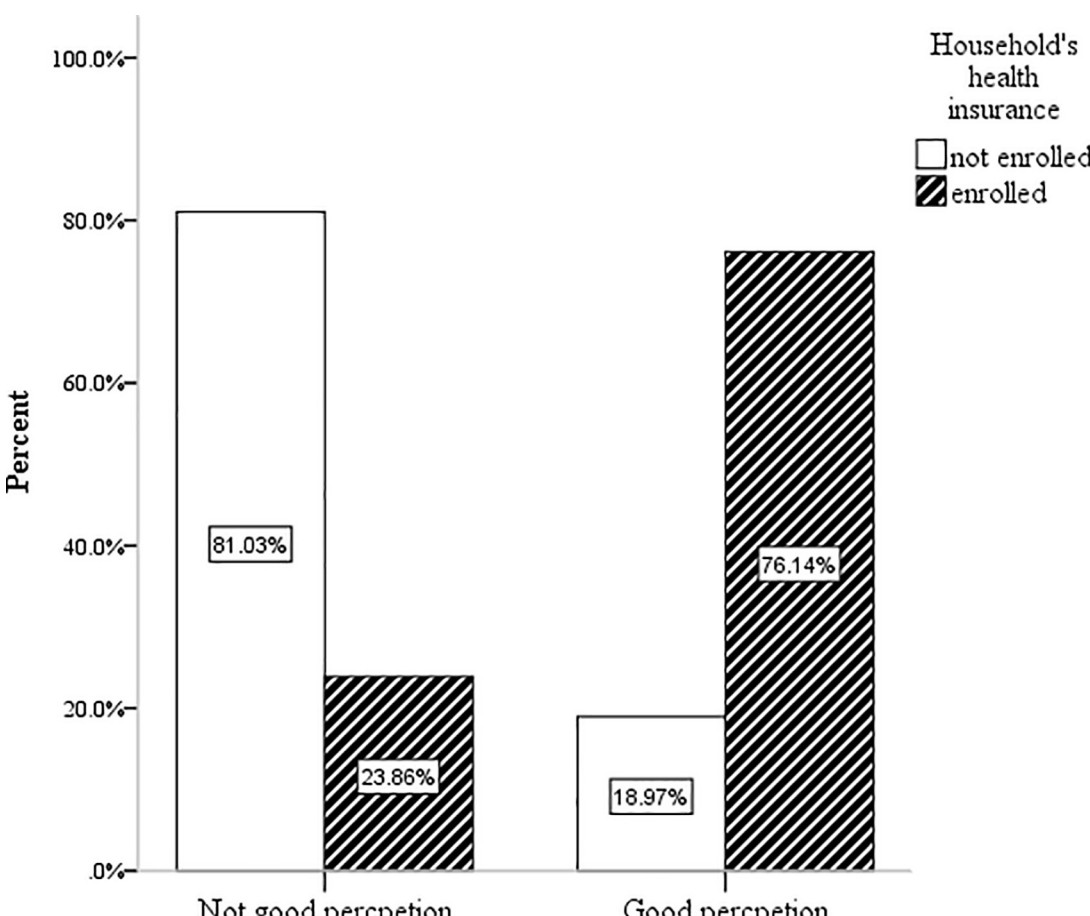

**Fig 1. Ranking of perception of participants towards enrolment of CBHI in Tach-Armachiho district, North Gondar, 2019.**

**Table 2. Bivariable and Multivariable logistic regression analysis of determinants of enrollment in CBHI among respondents, in Tach-Armachiho district, North Gondar, 2019.**

| Variables | Uptake of CBHI | | OR at 95% CI | |
|---|---|---|---|---|
| | Yes n (%) | No n (%) | COR (P-value) | AOR (95%CI) |
| **Sex of household head** | | | | |
| Female | 26 (29.5) | 30 (17.2) | 2.01(.023) | 2.79 (1.16, 6.69) * |
| Male | 62 (70.5) | 144 (82.8) | 1 | 1 |
| **Age of household head** | | | 1.08 (0.000) | 1.09 (1.05, 1.13) ** |
| **Religion of household head** | | | | |
| Orthodox | 27 (30.7) | 31 (17.8) | 1 | |
| Muslim | 61 (69.3) | 143 (82.2) | 0.49 (0.019) | |
| **Perception towards CBHI** | | | | |
| Negative perception | 21 (23.9%) | 141 (81%) | .073 (0.0000) | 0.062 (.030, .128)** |
| Positive perception | 67 (76.1%) | 33 (19%) | 1 | 1 |

*p-value < 0.05

**p-value < 0.01.

compared to female-headed households [11, 14, 16, 19]. However, a population-based case–control study conducted in rural Burkina Faso found that sex of the household head has no association with utilization of CBHI scheme [12]. This difference might be related to the different study settings, since different scheme design and countries experience affect groups differently.

Overall, this finding is an important landmark as it suggests more attention is needed to address heterogeneity of treatment effects by gender ensure accessibility of the scheme across the different segment of the population and as well, the need to involve women's in the decision-making process in the community in general and male heeded households in particular.

In this study, increase in age of household head was associated with more probability of using CBHI scheme. Also, this study was also consistent with studies conducted in Pilot Schemes found that, those with older household heads are more likely to enroll than with the opposite characteristics [7]. Similarly, studies conducted in Ghana, Mali, Senegal, Tanzania and Burkina Faso indicated that, increase in age of household head was positively associated with enrolment of CBHI [3, 20, 21]. This might be an increase in age could increase the probability of being insured to facilitate access to medical care for restoring decreasing health stock.

While, a population-based case–control study conducted in rural Burkina Faso found that age of the household head has no association with enrollment [12]. This difference might be related to the different study settings.

On top of that, this is an important finding as it suggests higher age groups joined the scheme, CBHI risks remaining an initiative exclusively accessible to certain groups within society; possiblly which raises concerns about scheme sustainability.

This study demonstrates that, household head's negative perception towards CBHI scheme factors (Joining the scheme will benefit me; CBHI management is trusted; quality of care is adequate and provider makes a good diagnosis) had the strongest association with enrolment in CBHI. The odds of household head's enrollment in CBHI was 16 times lower in household heads that had negative perception towards CBHI scheme.

This finding was consistence with study result conducted in Burkina Faso, Ghana and Nigeria found that, perceived CBHI beneficial in the dimension of effectiveness of CBHI benefit package; quality of care and trustworthiness of CBHI management as good are more likely to enroll than those who don't think as such [17, 18]. This might be the fact that, client expectations and/or previous experience related to health care service. On top of all, this is an important finding as it suggests that both positive and negative perceptions play an important role in household's enrolment in the scheme even though varying degree.

## Limitation of the study

Lack of similar studies particularly in case control design made difficult in comparing results and discuss some of the findings as needed.

## Conclusion

The study was conducted to identify determinants of enrollment in CBHI. The findings of the study indicated that, female-headed households and age were found in favoring enrollment while, negative perception towards CBHI scheme was found in discouraging enrolment.

At Regional Health Bureau, CBHI administrators, Zonal health office level and primary health care unit. Interventions done to improve scheme uptake should focus mainly on household heads that have negative perception towards CBHI scheme, male headed household and household heads age. Since more information is needed to explore the reason for low uptake

of CBHI, further study is recommend with the application of both quantitative and qualitative study design.

## Acknowledgments

First we would like to thank all study participants for their cooperation in providing the necessary information. We would also thank data collectors and supervisors for the devotion and quality work during data collection period.

## Author Contributions

**Conceptualization:** Muluken Genetu Chanie.

**Data curation:** Muluken Genetu Chanie, Gojjam Eshetie Ewunetie.

**Formal analysis:** Muluken Genetu Chanie, Gojjam Eshetie Ewunetie.

**Investigation:** Muluken Genetu Chanie, Gojjam Eshetie Ewunetie.

**Methodology:** Muluken Genetu Chanie, Gojjam Eshetie Ewunetie.

**Project administration:** Muluken Genetu Chanie.

**Resources:** Gojjam Eshetie Ewunetie.

**Software:** Muluken Genetu Chanie, Gojjam Eshetie Ewunetie.

**Supervision:** Muluken Genetu Chanie, Gojjam Eshetie Ewunetie.

**Validation:** Muluken Genetu Chanie, Gojjam Eshetie Ewunetie.

**Visualization:** Gojjam Eshetie Ewunetie.

**Writing – original draft:** Muluken Genetu Chanie, Gojjam Eshetie Ewunetie.

**Writing – review & editing:** Muluken Genetu Chanie.

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
