## [Decision Letter · Decision Letter 0]

19 Mar 2020

PONE-D-20-00655

Determinants of enrollment in Community based health insurance among Households in Tach-Armachiho Woreda, North Gondar, Ethiopia, 2019

PLOS ONE

Dear Dr. Chanie,

Thank you for submitting your manuscript to PLOS ONE. After careful consideration, we feel that it has merit but does not fully meet PLOS ONE’s publication criteria as it currently stands. Therefore, we invite you to submit a revised version of the manuscript that addresses the points raised during the review process.

We would appreciate receiving your revised manuscript by May 03 2020 11:59PM. To enhance the reproducibility of your results, we recommend that if applicable you deposit your laboratory protocols in protocols.io, where a protocol can be assigned its own identifier (DOI) such that it can be cited independently in the future. For instructions see: http://journals.plos.org/plosone/s/submission-guidelines#loc-laboratory-protocols

We look forward to receiving your revised manuscript.

Kind regards,

HASNAIN SEYED EHTESHAM

Academic Editor

PLOS ONE

Journal Requirements:

2. Please include additional information regarding the survey or questionnaire used in the study and ensure that you have provided sufficient details that others could replicate the analyses. If you developed and/or translated a questionnaire as part of this study and it is not under a copyright more restrictive than CC-BY, please include a copy, in both the original language and English, as Supporting Information.

3. We note that you have reported significance probabilities of 0 in places. Since p=0 is not strictly possible, please correct this to a more appropriate limit, eg 'p<0.0001

"Funding - University of Gondar"

"no"

6. Your ethics statement must appear in the Methods section of your manuscript. If your ethics statement is written in any section besides the Methods, please move it to the Methods section and delete it from any other section. Please also ensure that your ethics statement is included in your manuscript, as the ethics section of your online submission will not be published alongside your manuscript.

Additional Editor Comments (if provided):

Major Revision

Reviewers' comments:

Reviewer's Responses to Questions

**Comments to the Author**

1. Is the manuscript technically sound, and do the data support the conclusions?

Reviewer #1: Yes

Reviewer #2: Yes

2. Has the statistical analysis been performed appropriately and rigorously? 

Reviewer #1: Yes

Reviewer #2: Yes

3. Have the authors made all data underlying the findings in their manuscript fully available?

Reviewer #1: Yes

Reviewer #2: Yes

4. Is the manuscript presented in an intelligible fashion and written in standard English?

Reviewer #1: Yes

Reviewer #2: Yes

5. Review Comments to the Author

Reviewer #1: Manuscript #: PONE-D-20-00655

Title: Determinants of enrollment in Community based health insurance among Households in Tach-Armachiho Woreda, North Gondar, Ethiopia, 2019

Comments:

The aim of the present study is to identify factors affecting of enrollment in community based health insurance. The topic is out side the ambit of my subject expertise and an expert from the relevant field is required to do better justice to the article.

Reviewer #2: Comments to the authors:

The study by MUULKEN GENETU CHANIE and GOJJAM ESHETIE EWUNETIE identified determinants of enrollment in CBHI (community-based health insurance) among households in Tach-Armachiho district. The structured questionnaire revealed that, there was no significant association between marital-status, family size, educational status, distance, religion, and chronic illness in the uptake of CBHI. They showed that uptake of the scheme was significantly depends on the sex of the household head. They also demonstrated that female headed households were more likely to enroll as compared to male-headed households. These findings are similar to others conducted in Ghana, Mali and Senegal. Dissimilar to this, studies conducted in Fogera Woreda, North West Ethiopia, Burkina Faso and Nigeria found that male-headed households were more likely to enroll as compared to female headed households. There were few studies which founds that no association exist between household head and uptake of the scheme. The study also demonstrated the link between age of the head of the households, and uptake of the CBHI scheme. The manuscript is well written and presented. All of the important points are discussed in diligent manner. Few questions that can be discussed are given below:

1. Why the authors have chosen the different numbers of CBHI and non CBHI cases. The non CBHI cases are double to the CBHI cases? How this difference can change the outcome of the study?

2. Are the biased male cases in both CBHI and non CBHI case could influence the outcome?

3. The enrolled cases could be more including both CBHI and non CBHI cases. Are the authors think that this could influence the outcome of the study?

6. PLOS authors have the option to publish the peer review history of their article (what does this mean?). If published, this will include your full peer review and any attached files.

Reviewer #1: No

Reviewer #2: Yes: Mohd Shariq

---

## [Author Response · Author response to Decision Letter 0]

17 Jun 2020

2. Please include additional information regarding the survey or questionnaire used in the study and ensure that you have provided sufficient details that others could replicate the analyses. If you developed and/or translated a questionnaire as part of this study and it is not under a copyright more restrictive than CC-BY, please include a copy, in both the original language and English, as Supporting Information.

Dear editor thanks for your constructive comments. 

The survey tools were adopted from reviewing a couple of previous literatures. They were written in English language. They were paraphrased accordingly and referenced in the main document. No tool was developed by the authors.

---

## [Editor Report · Decision Letter 1]

29 Jun 2020

Determinants of enrollment in Community based health insurance among Households in Tach-Armachiho Woreda, North Gondar, Ethiopia, 2019

PONE-D-20-00655R1

Dear Dr. Chanie,

We’re pleased to inform you that your manuscript has been judged scientifically suitable for publication and will be formally accepted for publication once it meets all outstanding technical requirements.

Kind regards,

Hasnain Seyed Ehtesham

Academic Editor

PLOS ONE

Additional Editor Comments (optional):

I have gone through the revised manuscript and also the Authors response to reviewers comments. All comments of the reviewers have been satisfactorily addressed by the Authors. I recommend this manuscript for publication.
---

## [Editor Report · Acceptance letter]

23 Jul 2020

PONE-D-20-00655R1 

Determinants of enrollment in Community based health insurance among Households in Tach-Armachiho Woreda, North Gondar, Ethiopia, 2019 

Dear Dr. Chanie:

I'm pleased to inform you that your manuscript has been deemed suitable for publication in PLOS ONE. Congratulations! Your manuscript is now with our production department. 

Kind regards, 

on behalf of

Prof Hasnain Seyed Ehtesham 

Academic Editor

PLOS ONE